# Potentiation of Cisplatin Cytotoxicity in Resistant Ovarian Cancer SKOV3/Cisplatin Cells by Quercetin Pre-Treatment

**DOI:** 10.3390/ijms241310960

**Published:** 2023-06-30

**Authors:** Aseel Ali Hasan, Elena Kalinina, Julia Nuzhina, Yulia Volodina, Alexander Shtil, Victor Tatarskiy

**Affiliations:** 1T.T. Berezov Department of Biochemistry, RUDN University, 6 Miklukho-Maklaya St., 117198 Moscow, Russia; ali.aseel.hasan@gmail.com; 2Laboratory of Molecular Oncobiology, Institute of Gene Biology, Russian Academy of Sciences, 34/5 Vavilov Street, 119334 Moscow, Russia; julia.nuzhina@gmail.com (J.N.); tatarskii@gmail.com (V.T.); 3Center for Precision Genome Editing and Genetic Technologies for Biomedicine, Institute of Gene Biology, Russian Academy of Science, 34/5 Vavilov Street, 119334 Moscow, Russia; 4Laboratory of Tumor Cell Death, Blokhin National Medical Research Center of Oncology, 24 Kashirskoye Shosse, 115478 Moscow, Russia; uvo2003@mail.ru (Y.V.); shtilaa@yahoo.com (A.S.)

**Keywords:** quercetin, cisplatin, antioxidant systems, antioxidant enzymes, signaling pathway, pro-oxidant effect

## Abstract

Previously, we demonstrated that the overexpression of antioxidant enzymes (*SOD-1*, *SOD-2*, *Gpx-1*, *CAT*, and *HO-1*), transcription factor *NFE2L2*, and the signaling pathway (*PI3K/Akt/mTOR*) contribute to the cisplatin resistance of SKOV-3/CDDP ovarian cells, and treatment with quercetin (QU) alone has been shown to inhibit the expression of these genes. The aim of this study was to expand the previous data by examining the efficiency of reversing cisplatin resistance and investigating the underlying mechanism of pre-treatment with QU followed by cisplatin in the same ovarian cancer cells. The pre-incubation of SKOV-3/CDDP cells with quercetin at an optimum dose prior to treatment with cisplatin exhibited a significant cytotoxic effect. Furthermore, a long incubation with only QU for 48 h caused cell cycle arrest at the G1/S phase, while a QU pre-treatment induced sub-G1 phase cell accumulation (apoptosis) in a time-dependent manner. An in-depth study of the mechanism of the actions revealed that QU pre-treatment acted as a pro-oxidant that induced ROS production by inhibiting the thioredoxin antioxidant system Trx/TrxR. Moreover, QU pre-treatment showed activation of the mitochondrial apoptotic pathway (cleaved caspases 9, 7, and 3 and cleaved PARP) through downregulation of the signaling pathway (mTOR/STAT3) in SKOV-3/CDDP cells. This study provides further new data for the mechanism by which the QU pre-treatment re-sensitizes SKOV-3/CDDP cells to cisplatin.

## 1. Introduction

Despite the significant strides that have been made towards the development of targeted anticancer therapies, chemotherapy remains the first-line treatment for most cancers [1,2]. Platinum-based antitumor drugs, such as cisplatin (CDDP) (Figure 1), are established chemotherapeutic agents in the treatment of a number of cancers and sarcomas; however, the development of cisplatin resistance is a major obstacle to curative cancer treatment [3]. Drug resistance phenomena can be developed through several mechanisms, such as the overexpression of drug efflux pumps, activation of DNA repair efficiency, escape of drug-induced apoptosis, active cell survival signals, and enhanced drug detoxifying systems [4,5].

Under normal physiological conditions, the cellular redox homeostasis ensures that the cells respond appropriately to endogenous and exogenous stimuli. The redox systems are perfectly suited to the regulation of the cell’s life-or-death decisions [6]. Both antioxidant enzymes (superoxide dismutase (SOD), glutathione reductase (GR), catalase (CAT), etc.) and low molecular weight antioxidants are, effectively, inbuilt defense strategies to maintain cellular redox homeostasis during oxidative stress [7], which is defined as a disturbance in the balance between the generation of reactive oxygen and nitrogen species (RONS) and the antioxidant systems in the cell. The electron transport chain (ETC) in mitochondria and the NADPH oxidases (NOX) are the major endogenous sources of ROS, including the superoxide anion (O_2_^•–^), hydrogen peroxide (H_2_O_2_), organic hydroperoxides (ROOH), the hydroxyl radical (^•^OH), and the peroxyl radical (ROO), inside the cell. Enzymatic antioxidant defense for ROS elimination occurs as a series of the following reactions: SODs catalyze the dismutation of the O_2_^•–^ into H_2_O_2_. GPx catalyzes the detoxification of H_2_O_2_ by reduced glutathione (GSH). Further on, GR reduces oxidized glutathione (GSSG) to (GSH) using NADPH as the electron donor, while H_2_O_2_, which is found mainly in peroxisomes and cytosol, is neutralized by the CAT enzyme [8].

In addition to its role in defense against oxidative stress, the thioredoxin (Trx) system also regulates DNA synthesis, the cell cycle, and the apoptosis process in mammalian cells [9]. The over-activated Trx system, which contributes significantly to cancer progression and therapy resistance, has been recognized in several human tumors, including such highly aggressive types as lung, liver, pancreas, ovarian, and breast cancers. This antioxidant system is composed of cytosolic/nuclear and mitochondrial thioredoxin reductases (TrxR-1 and TrxR-2) and thioredoxins (Trx-1 and Trx-2), respectively. TrxR-1 and TrxR-2 are able to transfer electrons from NADPH to oxidized forms of Trx-1 and Trx-2, respectively, which reduce their substrates using a highly conserved dithiol form of the active site sequence [8]. Oxidized signal transducers and activators of transcription 3 (STAT3) dimers, being one of the Trx target proteins, receive electrons from a reduced Trx form to regenerate reduced STAT3 monomers. Upon Tyr705 phosphorylation, the transcriptionally active phosphorylated STAT3 (pSTAT3) homodimer is formed, which in turn translocates to the nucleus and interacts with specific DNA sequences to promote STAT3-dependent gene expression [10]. The accumulating evidence shows that the overexpression of STAT3, a transcription factor, has been observed in several cancer types and that it correlates with increased tumor cell proliferation, survival, self-renewal, tumor invasion, and angiogenesis, as well as higher cellular resistance to cisplatin [11,12]. Another important signaling pathway that correlates with pro-survival, the mechanistic target of rapamycin (mTOR) signaling, is also associated with CDDP resistance; it is necessary to phosphorylate STAT3 on serine^727^ to ensure its maximum activation [11].

A number of epidemiological studies have validated the fact that the consumption of dietary polyphenols, mainly quercetin (QU) (Figure 1), is inversely correlated with cancer incidence; this has been attributed to the antioxidant properties of QU, which prevent cancer through the increasing expression of antioxidant enzymes [13]. Moreover, QU exerts its anticancer activity by modulating cell cycle progression, reducing cell proliferation, and inhibiting angiogenesis and metastasis progression [14], as well as by promoting apoptosis and autophagy via the modification of various pathways, such as the PI3K/Akt/mTOR, Wnt/β-catenin, and MAPK/ERK1/2 pathways [15]. QU, a DNA intercalator, causes S-phase arrest during cell cycle progression and is consistent with the plausible induction of DNA damage following DNA intercalation within the cancer cells, which can eventually lead to the inducing of apoptosis [16]. In addition to these mechanisms, QU can also trigger apoptosis via various mechanisms that depend on the type of cancer cells. For instance, QU inhibited the growth of human A375SM melanoma cells through inducing apoptosis via activation of the JNK/P38 MAPK signaling pathway [17]. In vitro and in vivo mitochondrial-derived apoptosis following QU treatment in HL-60 AML cells was found to occur via reactive-oxygen-species-mediated ERK activation [18].

However, the clinical application of QU is limited due to its poor solubility, low bioavailability, poor permeability, and instability [19]. The bioavailability of quercetin can be significantly enhanced when it is consumed as an integral food component [20]. The bioavailability of QU can be improved by encapsulating it in polymer micelles. The intravenous administration of QU encapsulated in biodegradable monomethoxy poly (ethylene glycol)-poly (ε-caprolactone) (MPEG-PCL) micelles significantly inhibited the growth of established xenograft A2780S ovarian tumors through the activation of apoptosis and the inhibition of angiogenesis in vivo. This confirms the effectiveness of the clinical application of QU in the treatment of ovarian cancer [21].

QU also showed an anticancer effect on chemotherapy-resistant cells, such as CDDP, through the inhibition of the proliferation of ovarian carcinoma (SKOV3) and osteosarcoma (U2OS) human cell lines, as well as their cisplatin (CDDP)-resistant counterparts. Furthermore, the inhibition of the cyclin D1 level was associated with G1/S-phase alteration in QU-treated cells. CDK and cyclin inhibition by QU may be a viable therapeutic target in the ovarian CDDP-resistant cell line. Due to the ability of quercetin to overcome the resistance of cancers toward CDDP, a significant emphasis should be placed on using combinatory chemotherapy with cytotoxic drugs, particularly CDDP [22].

Despite the several studies which have investigated the anticancer effects of QU in in vitro and in vivo models, some of these effects have not been observed in ovarian cancer. However, to the best of our knowledge, the underlying mechanisms by which QU sensitizes ovarian cancer cells to CDDP, particularly those with acquired CDDP resistance, e.g., SKOV-3/CDDP, remain elusive and still need to be elucidated to a large extent. In the previous study, after treatment with QU alone, cisplatin-resistant SKOV-3/CDDP ovarian cancer cell lines exhibited inhibition in the gene expressions of antioxidant enzymes (*SOD-1, SOD-2, Gpx-1, CAT*, and *HO-1*), transcription factor *NFE2L2*, and the signaling pathway (*PI3K/Akt/mTOR*), which contribute to cisplatin-resistance in these cells [23].

In the present study, we attempted to explore the anticancer effects of QU pre-treatment on the CDDP-resistant ovarian adenocarcinoma human SKOV-3/CDDP cell line model. For the first time, we provide evidence that QU pre-treatment exerts pro-oxidant activity on the SKOV-3/CDDP cell line via alleviation of the protein expression of the cytosolic and mitochondrial thioredoxin antioxidant (Trx/TrxR) system that maintains STAT3 in a reduced/active state. We also identified the mechanism by which QU pre-treatment sensitizes SKOV-3/CDDP cells to cisplatin-induced apoptosis through a downregulation signaling (mTOR/STAT3) pathway. Thus, this research provides a better understanding of the mechanisms involved in QU-mediated CDDP sensitization in SKOV-3/CDDP ovarian cancer cells.

## 2. Results

### 2.1. QU Pre-Treatment Inhibits the Cell Survival of SKOV3 and SKVO3/CDDP Cell Lines

In our recently published research articles [23,24], IC_50_ doses of the QU and CDDP under investigation in the SKOV3 human ovarian cancer cell line and the CDDP-resistant subline, SKVO3/CDDP, were screened as cell models. In order to determine whether QU pre-treatment impairs the CDDP resistance in the SKOV3/CDDP cell line, we designed an alternative QU pre-treatment strategy to compare it to the classical individual QU or CDDP treatment, assuming that the individual treatment with QU or CDDP does not exhibit an effect; the significant effect appears only when co-treatment is applied (Figure 2a).

Based on our data, we decided to use the optimum minimum and maximum effective doses of QU and CDDP as the reference values for QU (15 and 100 µM) for 24 h [23] (Appendix A). However, we found that a long pre-incubation with a significantly non-toxic moderate dose (60 µM) of QU for 48 h was recognized as having an optimum effect on cell viability, enabling it to re-sensitize the SKOV-3/CDDP cisplatin-resistant human ovarian adenocarcinoma subline to CDDP, which was used in this experiment and further on in the rest of the experiments in this study. For instance, at 60 µM of QU for 48 h, followed by 5 µM (1/2 IC_50_) for 72 h, CDDP significantly resulted in the inhibition (65% decrease; *p* < 0.0001) of the viability of the SKOV-3 cells, compared with CDDP (5 µM; 20% decrease) and QU (60 µM; 35% decrease) alone (Figure 2b).

However, in the SKOV-3/CDDP subline cells, CDDP at the doses of 5 µM and 10 µM did not result in any loss of viability, suggesting that these cells are endogenously resistant to CDDP compared to the SKOV-3 cells (Appendix A). The MTT assay revealed that the percentage of QU-treated cells decreased concomitantly when cisplatin was used at the dose of 34 μM (IC_50_) and at 17 μM (1/2 IC_50_) (Appendix A). Thus, for the SKOV-3/CDDP subline, the non-toxic dose of CDDP was optimized at 17 µM (1/2 IC_50_). In a similar manner to the SKOV-3 cells, the pre-incubation of SKOV-3/CDDP subline cells with QU significantly resulted in the inhibition of cell survival (65% decrease at 17 µM concentration of CDDP for 72 h; *p* < 0.005), compared with the treatments with only CDDP (17 µM; 20% decrease) and QU (60 µM; 35% decrease) alone (Figure 2b). These data revealed that the pre-treatment with QU followed by CDDP effectively induced survival inhibition in both ovarian cancer cells.

### 2.2. QU Pre-Treatment Increases Sub-G1 Phase Cell Accumulation in SKOV3 and SKVO3/CDDP Cell Lines, with Increasing Apoptotic Cell Percentages

As cytotoxicity is frequently accompanied by cell cycle arrest, the impact of QU or CDDP alone and QU pre-treatment in SKOV-3 and SKOV-3/CDDP cells was evaluated by cell DNA content using PI staining for flow cytometry (Figure 3). The cell cycle analysis showed that at the lower dose of 5 µM (1/2 IC_50_ SKOV-3) of CDDP treatment, the SKOV-3 cells were arrested at the S phase (Figure 3a), while only at the higher dose of 17 µM (1/2 IC_50_ SKOV-3/CDDP) of CDDP were the SKOV-3/CDDP cells arrested at the S and G2/M phases and slightly accumulated at the sub-G1 phase in a time-dependent manner (Figure 3b). Compared to the control groups, the sub-G1 and G2/M phases of the SKOV-3 and SKOV-3/CDDP cells treated with only QU were time-dependently increased (Figure 3). Therefore, the effect of CDDP on cell cycle distribution in QU pre-treated cells was the induction of sub-G1 accumulation, which indicated apoptosis-associated chromatin degradation and the arrest of the cell cycle in the G2/M or S phases. For the SKOV-3 cells, the QU pre-treatment resulted in a significant increase in the sub-G1 phase, where it increased from 61 ± 1% at 24 h to 72.4 ± 2% at 48 h (Figure 3c). Meanwhile, in comparison, the percentage of cells in the sub-G1 phase at 72 h for the SKOV-3/CDDP cells was 53.5 ± 0.7% (Figure 3d). In our previous study [23], we revealed that the treatment of cells with only QU at a high dose (100 µM for 24 h) slightly increased the population of ovarian treated cells in the S and sub-G1 phases, while with a long incubation period, mainly of 48 h, QU significantly arrested the cell cycle at the G1/S phase and at the same time had no toxicity effect even after 72 h of incubation (Appendix A). Representative flow cytometry images for the cell cycle analysis are shown in (Figure 3).

Here, in addition to the pre-treatment strategy, we studied the synergistic approach by adding QU and CDDP at the same time for 24 h, 48 h, and 72 h and analyzed the effect of this combination on the cell cycle. Unexpectedly, the results of the cell cycle showed that the synergistic approach in SKOV-3/CDDP leads to an arrested cell cycle in the G2/M or S phases without a significant increase in the sub-G1 phase, even after the long incubation (Appendix A). On the other hand, QU pre-treatment significantly increased the percentage of cells in the sub-G1 phase and the sensitivity of the cisplatin-resistant ovarian cancer cells SKOV-3/CDDP to CDDP even after replacing QU with a fresh medium for a period of 24, 48, and 72 h.

### 2.3. QU Pre-Treatment Induces Cellular Reactive Oxygen Species (ROS) Production by CDDP in SKVO3/CDDP Cell Line

To investigate whether the accumulation of both ovarian cancer cells in sub-G1 by QU pre-treatment was due to the induction of oxidative stress, intracellular ROS production was evaluated by using the CellROX Deep Red assay (Figure 4). Our study suggested that QU pre-treatment could induce earlier ROS production; therefore, we investigated the change in the ROS level in pre-treated ovarian cancer cells. The treatment with QU alone for 24 h and 48 h was found to reduce the ROS levels in comparison to those of the control group in both types of cells (Figure 4). Following CDDP exposure for 4 h, no increase in ROS levels was observed in either the SKOV-3 or the SKOV-3/CDDP cells (Figure 4). In SKOV-3, after 24 h, CDDP triggered the ROS generation up to four-fold (17 ± 2.9% cells against 4 ± 0.3% in the control), whereas in the SKOV3/CDDP cells it was only up to three-fold; this can be explained by the high level of antioxidant system in the resistant cells [23] (Figure 4b). The QU pre-treatment reduced the effect of CDDP through decreasing the intracellular ROS to the control level in SKOV-3 (Figure 4b). Conversely, in SKOV-3/CDDP, the proportion of the fluorescent cells was 5 ± 1% and 9 ± 1% in the untreated cells and the CDDP group, respectively, whereas the QU pre-treated cells had 23 ± 1% fluorescent cells (Figure 4b).

According to these results, we evaluated the increase in the pro-oxidant effect of QU pre-treatment concomitant with ROS production in the SKOV-3/CDDP cells. Interestingly, this polyphenol compound had an antioxidant effect at 24 and 48 h and a pro-oxidant effect only in the pre-treatment group. These results provide support for the proposal that QU pre-treatment increased the sensitivity of CDDP-resistant ovarian cancer SKOV-3/CDDP cells to CDDP and induced the accumulation of SKOV-3/CDDP cells in sub-G1 by promoting ROS formation.

### 2.4. QU Pre-Treatment Alleviates Protein Expression of Thioredoxin *Antioxidant* System Trx/TrxR, as Well as the mTOR/STAT3 Signaling Pathway in SKVO3/CDDP Cell Line

To further elucidate whether the inducing of the accumulation of SKOV-3/CDDP cells in sub-G1 and ROS production by QU pre-treatment is mediated through modulation of the antioxidant system and signaling pathway, the protein expression level associated with sensitivity to CDDP was assessed in ovarian cells using Western blot analysis. Our study identified that the expressions of the important thioredoxin antioxidant system Trx/TrxR, as well as the mTOR/STAT3 signaling pathway, were dramatically higher in the SKOV-3/CDDP subline cells compared with those in the SKOV-3 cells (Figure 5a–c). This suggests that the overexpression of these proteins makes a significant contribution to the development of SKOV-3/CDDP cells in their acquiring of resistance to CDDP. Previously, we used SKOV-3 and SKOV-3/CDDP subline cells to analyze the effect of two different doses (15 µM and 100 µM) of QU in modulating antioxidant enzymes (*SOD-1*, *SOD-2*, *Gpx-1*, *CAT*, and *HO-1*), transcription factor *NFE2L2*, and the protein kinase pathway (*PI3K/Akt/mTOR*) gene expression levels [23].

In the present experiment, we analyzed the most effective dose of QU followed by CDDP in modulating the antioxidant system and protein kinase signaling pathway proteins. Compared with individual action, the highest inhibition effect in SKOV-3/CDDP subline cells that led to their increased sensitivity to CDDP was observed in the protein expression levels after incubation with 60 µM QU for 48 h, followed by 5 µM CDDP for 24 h in SKOV-3 and 17 µM CDDP for 48 h in SKOV-3/CDDP (Figure 5a,b,d,e). The expression levels of the cytosolic and mitochondrial Trx/TrxR system and the active phosphorylated forms of the mTOR/STAT3 proteins in the SKOV-3/CDDP subline cells were effectively decreased by the QU pre-treatment.

### 2.5. QU Pre-Treatment Triggers Apoptotic Mitochondrial Pathway in SKOV3 and SKVO3/CDDP Cell Lines

To determine whether the growth inhibition and accumulation of both ovarian cancer cells in sub-G1 by pre-treatment with QU were due to the induction of apoptosis, cell apoptosis was assessed by flow cytometry based on Annexin-V/PI double staining analysis. In contrast to the control group, a single treatment of CDDP mildly induced apoptosis at an average of 0. 9 ± 0.1 and 4 ± 0.6% in the SKOV-3 cells and at an average of 2.5 ± 0.4 and 2 ± 0.1% in the SKOV-3/CDDP cells for 24 h and 48 h, respectively. QU monotherapy slightly reduced the live cells and slightly increased apoptosis to an average of ~3 ± 0.4% and ~1.5 ± 0.4% in SKOV-3 and SKOV-3/CDDP, respectively, compared to the untreated cells (Figure 6a,b).

Apoptosis was induced by QU pre-treatment, which induced necrotic cells in a time-dependent manner (Figure 6a,b). As the incubation time increased, the proportions of Annexin V+/PI− (early stage of apoptosis) and Annexin V+/PI+ (late stage of apoptosis) QU pre-treated ovarian cells were significantly higher than the CDDP and QU groups (Figure 6a,b); in particular, apoptosis was induced by approximately 20.3 ± 3% in the SKOV-3 QU pre-treated cells after 48 h, compared with the single-drug treatment group (Figure 6a). The results also indicated that QU sensitization prior to treatment with CDDP in the SKOV-3/CDDP cells substantially increased the percentage of apoptosis by ~41.5 ± 2% as compared to the treatment with CDDP, with only 2 ± 0.1% apoptosis after 48 h (Figure 6b). 

To further detect the mechanisms behind the apoptotic process induced by treatments using either QU or CDDP and QU pre-treatment, we investigated the expression of specific apoptosis marker proteins (cleaved caspase-9,7,3 and cleaved PARP) with Western blot in both cancer cells. Compared with the loading control histone protein, the levels of cleaved caspase-9, -7, and -3 proteins in both cells were observed only in the QU-pre-treated cells, while the QU pre-treatment induced a high expression of cleaved PARP in both cell lines compared with the mono-treatments (Figure 6c). In other words, these results demonstrated that QU pre-treatment effectively inhibited cell growth by increasing apoptosis in ovarian cancer cells. Taken together, the cell death effect of the studied QU compound was mediated by inhibition of the mTOR/STAT3 pathway, as well as stimulation of the ROS-mediated caspase-3, -7, and -9 activations.

## 3. Discussion

Ovarian cancer is a continuing health problem that accounts for a significant portion of mortality among women around the world [25,26]. In current clinical practice, surgery combined with chemotherapy based on platinum compounds is considered the standard therapeutic strategy for ovarian cancer [27]. CDDP is one of the most effective chemotherapeutic agents for ovarian cancer; however, resistance to this chemotherapeutic drug is a major obstacle to curative cancer therapy. Ovarian cancer cells develop resistance to anticancer drugs through various mechanisms, including DNA damage repair, cell metabolism, oxidative stress, cell cycle regulation, apoptotic pathways, and abnormal signaling pathways [28]. Due to their various beneficial properties, polyphenols are potential therapeutic candidates in cancer when combined with classical antitumor agents such as cisplatin [29]. For instance, QU could prevent ovarian cancer with its anti-inflammatory, pro-oxidative, and antiproliferation effects, and cell cycle arrest [26]. Discovering the main quercetin mechanisms that reduce cisplatin resistance is important for improving cancer treatment. However, the role of quercetin in SKOV-3/CDDP has not yet been completely investigated. 

We initially focused on the effect of QU alone on the ovarian adenocarcinoma SKOV-3/CDDP cell lines to establish a baseline for comparison with our later study [23]. Hence, we hypothesized that the pre-exposure to QU might impair cisplatin resistance in ovarian adenocarcinoma cell lines. In the present study, we utilized two adenocarcinoma cell lines, i.e., the SKOV-3 (wild type) and SKOV-3/CDDP cisplatin-resistant sublines. Optimum doses for treatment were decided for QU and CDDP after examining the IC_50_ values from our previously published data [23,24]. Pre-treatment with QU at low (15 µM) and high doses (100 µM) for 24 h followed by CDDP showed an antiproliferative effect in the SKOV-3/CDDP cisplatin-resistant subline, as well as in the SKOV-3 wild cells, and it showed a sensitization of the resistant subline to the cytotoxicity of CDDP (Appendix A), while a long incubation for 48 h showed a maximum antiproliferative effect (Figure 2b). 

The pre-treatment of leiomyosarcoma cells with epigallocatechin-3-gallate, a polyphenol compound, for 24 h failed to alter the S-phase cell cycle arrest induced by CDDP and to modulate the CDDP effects on mitochondrial function [30]. Furthermore, a recent study revealed that the higher efficacy of the prolonged pemetrexed pre-treatment of malignant pleural mesothelioma (MPM) for 48 h induced a cell cycle arrest, mainly in the G2/M phase, the accumulation of persistent DNA damage, and the induction of a senescence phenotype, thereby sensitizing cancer cells to subsequent CDDP treatment [31]. In our study, the treatment of ovarian cancer cells with only QU for 24 h was shown to lead to a slightly increasing cell cycle in the S phase, whereas a long treatment for 48 h significantly induced a cell cycle arrest in the G1/S phase (Appendix A). The conditions of a long treatment for 48 h significantly induced the cells to accumulate in sub-G1 after treatment with CDDP and increased the sensitization of the SKOV-3/CDDP subline to the subsequent CDDP treatment (Figure 3b,d). The sensitization of the ovarian cells to CDDP was observed when the polyphenols, curcumin or QU, were added synergistically with CDDP, as well as when they were added 24 h before [32]. In this study, two strategies (synergistic and pre-treatment) were designed. The result revealed that the SKOV-3/CDDP cells failed to enter into the sub-G1 phase even after a long incubation using the synergistic approach, while the cells effectively accumulated in sub-G1 using the QU pre-treatment strategy (Appendix A). Our strategy allowed the progression of the QU-treated cells into the S phase, whereby re-exposure of the QU-exhausted cells to CDDP after washing out the QU from the culture medium was sufficient to increase the duration of the S phase, which in turn led to further cell damage. The evaluation of this pre-treatment therapy in cisplatin-resistant ovarian cancer cells may lead to more efficient CDDP treatment.

The antioxidant or pro-oxidant properties of polyphenol compounds depend predominantly on the number and positions of the substituent hydroxyl groups, along with their redox metal (Cu, Fe) chelating capacity. For example, the higher number of hydroxyl groups, including the 3-OH group of the C ring of polyphenols in the presence of Cu(II) ions, act as potential anticancer agents with moderate pro-oxidant activity, causing DNA damage via interaction with DNA and ROS formation via the Fenton reaction. Otherwise, the slight pro-oxidant activity of polyphenols acts as a preventive anticancer therapeutic agent via the inducing of cellular antioxidant systems, including antioxidant enzymes, and the synthesis of low-molecular-weight antioxidants, such as glutathione [33]. In this study, QU could not effectively induce ROS generation, while CDDP slightly induced the ROS level in the SKOV/CDDP cells. However, QU pre-treatment effectively triggered the generation of ROS in the SKOV-3/CDDP cells. These data indicate that the increased oxidative stress induced by QU pre-treatment may have activated an apoptotic cascade in the SKOV-3/CDDP cells (Figure 4) and inhibited expression of antioxidant proteins and the signaling pathway (Figure 5a,b,d,e). Alternatively, QU pre-treatment in SKOV-3 acts as an antioxidant by decreasing ROS formation and inducing the expression of antioxidant proteins (Figure 5a,d) and the signaling of the mTOR/STAT3 pathway (Figure 5b,e). In the SKOV-3/CDDP cells, the oxidative stress was also monitored by the measuring of the antioxidant system and the signaling pathway at the protein levels.

In addition to altered proliferation, the cell cycle, oxidative stress, cell metabolism, increased ability to repair DNA damage, and reduced susceptibility to apoptosis, ovarian cancer cells develop resistance to anticancer drugs through various other mechanisms, defined as enhancement expression and alteration of signaling pathways [28]. In addition to the generation of ROS species, we also monitored the changes in the oxidative stress response proteins. Therefore, the marker proteins that were highly correlated with cellular sensitivity to CDDP were screened. Notably, we previously [23] provided the molecular evidence that antioxidant enzymes (*SOD-1, SOD-2, Gpx-1, CAT*, and *HO-1*), transcription factor *NFE2L2*, and the PI3K/Akt/mTOR signaling pathway are upregulated in SKOV-3/CDDP subline cells compared to normal line SKOV-3 cells and found that these gene expressions were associated with CDDP tolerance. The individual treatment with QU revealed marked downregulation in the expression of these genes. Our results suggest that QU pre-treatment has the potential to increase sensitivity to CDDP in SKOV-3/CDDP subline cells by regulating oxidative stress (ROS generation) and cell apoptosis via the antioxidant system and the signaling pathway. 

Hence, we assessed the pro-oxidative therapeutic potential effect of QU pre-treatment on the antioxidative system. The thioredoxin system (Trx/TrxR) is essential to maintaining cellular redox homeostasis in living organisms through detoxification of harmful metabolites, i.e., ROS; however, the overexpression of the Trx-dependent system can also contribute to the intensifying of the process of oncogenesis, such as by the enhancement of tumor growth, angiogenesis, and resistance to therapy via the modulation of both the gene expression and the cell signaling pathways that lead to the regulation of apoptotic pathways in cancer cells [34]. Compared to SKOV-3, which exhibits normal Trx/TrxR protein expressions, SKOV-3/CDDP subline cells display unique overexpression in both the cytosolic (TrxR-1) and mitochondrial (Trx-2) protein levels, which are directly associated with resistance to cisplatin (Figure 5c). Accumulating evidence suggests that the tumoral activity of the Trx-dependent system can be modulated by natural polyphenolic compounds, which act as Trx/TrxR system inhibitors [34]. The inhibitory effect of QU depends on many factors, including concentration, NADPH, and the time of exposition and involvement in an attack on the reduced COOH-terminal active site sequence -Gly-Cys-Sec-Gly of TrxR. The inhibition of TrxR activity is associated with the oxidization of Trx in the cells [35]. As expected, QU pre-treatment effectively reduced the levels of thioredoxin system (Trx-1, TrxR-1, and Trx-2) proteins in the SKOV-3/CDDP cells (Figure 5a,d).

Furthermore, STAT3 and its two phosphorylated forms (on Ser^727^ and Tyr^705^) were also found to be inhibited at the protein (Figure 5b,e) level by the QU pre-treatment in the SKOV-3/CDDP subline cells. Several TrxR-1 inhibitors also block the signal transducers and activators of transcription 3 (STAT3) activity via accumulation of oxidized STAT3, which blocks STAT3-dependent transcription [10,36] and induces cancer cell death [10]. Due to the thiol groups in the Cys residues in the active sites, reduced Trx catalyzes the reduction in disulfide bonds within oxidized target proteins [36,37], such as STAT3 [36].

Previously, SKOV-3/CDDP subline cells were shown to overexpress the *mTOR* gene, which was downregulated after treatment with QU [23]. A recent study demonstrated a significant correlation between STAT3 and mTOR overexpression, which underlies cancer-mediated drug resistance and cancer progression. Furthermore, the phosphorylation of STAT3 on Ser^727^ is regulated by mTOR [11]. Our data support the findings of the study; in particular, both STAT3 and mTOR were overexpressed in SKOV-3/CDDP compared with SKOV-3, and after QU pre-treatment, SKOV-3/CDDP showed inhibition of two phosphorylated STAT3 isoforms at Ser^727^ and Tyr^705^, as well as the inhibition of phospho-mTOR Ser^2448^. This was followed by mTOR/STAT3 pathway inactivation, which ultimately inhibits proliferation and induces caspase-dependent apoptosis in ovarian SKOV-3/CDDP cancer cells. 

Apoptotic cell death can be triggered by inhibition of the mTOR/STAT3 signaling pathways that are closely linked with tumor progression. Moreover, inducing cell cycle arrest by enhancing the p53 phosphorylation and apoptosis pathway after treatment with QU in HPV-positive human cervical-cancer-derived cells may be one of the key mechanisms underlying the inhibition of cell proliferation in cancer cells [38]. Annexin V/PI staining showing early and late apoptotic cells suggests the activation of apoptosis following pre-exposure to QU (Figure 6a,b). The results of this study also indicated that QU pre-treatment induces SKOV-3 and SKOV-3/CDDP cell death through the mitochondria-dependent apoptosis pathway, where mitochondrial dysfunction, increased PARP cleavage, and activation of caspase-9,-7,-3 are critical events for apoptosis (Figure 6c). QU showed induction in both necrosis and apoptosis cell death in various cancer cells, including SCC-9 oral cancer cells [39] and prostate cancer over a period of time [40]. Apoptosis was induced by QU pre-treatment, and necrotic cells were induced in a time-dependent manner (Figure 6a,b). The proposed anti-cancer and sensitizing mechanisms of QU pre-treatment are summarized in Figure 7.

Taken together, the anticancer and chemosensitizing effects of QU pre-treatment are associated with the inducing of ROS-mediated apoptosis and the alleviation of the protein expressions that are highly correlated with cellular sensitivity to cisplatin, including antioxidant enzymes and the signaling pathway.

## 4. Materials and Methods

### 4.1. Reagents

The chemicals used in the experiments were purchased from the following suppliers: cisplatin (cis-platinum (ii)-diamine dichloride (CDDP; Teva, Tel Aviv-Yafo, Israel)); 3-(4,5-dimethylthiazol-2-yl)-2,5-diphenyl tetrazolium bromide (MTT) (PanEco); and quercetin (QU) (3,3′,4′,5,7-pentahydroxyflavone) from (Acros Organics, Geel, Belgium). The chemical structures of CDDP and QU are illustrated in (Figure 1). The purity of QU was [97% (HPLC)]. The stock quercetin solution was immediately prepared before use by dissolving these test agents in dimethyl sulfoxide (DMSO) at 100 mM concentration.

### 4.2. Cell Culture

The SKOV-3 non-resistant human ovarian adenocarcinoma and SKOV-3/CDDP cisplatin-resistant subline cells were obtained from the All-Russian Scientific Center for Molecular Diagnostics and Treatment. The SKOV-3 and SKOV-3/CDDP cells were cultured in DMEM cell culture media supplemented with 10% heat-inactivated fetal bovine serum (HealthCare, Chicago, IL, USA), 2 mM L-glutamine, 100 U/mL penicillin, and 50 μg/mL streptomycin. All the cells were incubated at 37 °C under 5% CO^2^ in a humidified incubator.

### 4.3. Cell Viability (MTT Assay)

The inhibitory effect of the pre-treatment of QU followed by CDDP on ovarian cancer cells was assessed using the MTT assay. The SKOV-3 and SKOV-3/CDDP cells were seeded into 96-well plates (Nunc, ThermoFisher Scientific, Waltham, MA, USA) at a density of 5 × 10^3^ cells per well overnight in triplicates. After the incubation period with QU, the culture medium was replaced with a fresh medium, and the cells were then treated with (5 µM, 1/2 IC_50_, for SKOV-3) or (17 µM, 1/2 IC_50_, for SKOV-3/CDDP) of CDDP for 72 h. The culture medium was again replaced with a fresh medium and 20 μL; 5 mg/mL MTT was added to each well; and the plates were further incubated at 37 °C for 4 h, allowing the viable cells to reduce the yellow MTT to dark blue formazan crystals. The supernatant was removed, and the formazan crystals were dissolved in 100 μL DMSO and mixed thoroughly prior to determination of the absorbance at a wavelength of 570 nm using a Multiscan FC plate reader (Thermo Scientific, Waltham, MA, USA). Cell viability was calculated according to the following equation: cell viability (%) = (absorbance of experiment group/absorbance of control group) × 100. The IC_50_ was developed by an inhibition curve and recorded as the mean ± standard deviation of three independent experiments.

### 4.4. Cell Cycle Assay

The distribution of the cell cycle was analyzed by flow cytometry using the FACS Canto II (BD Biosciences, Franklin Lakes, NJ, USA). The SKOV-3 and SKOV-3/CDDP cells in the logarithmic growth phase were seeded at a density of 1.5 × 10^5^ cells/well in a 6-well plate, incubated at 37 °C, 5% CO_2_ until adherent, and treated with CDDP alone and QU alone or followed by CDDP, as described above. After incubation, the cells were harvested and washed once with phosphate-buffered saline (PBS). The pellets were then lysed in the solution containing 0.1% sodium citrate, 0.3% NP-40, 50 μg/mL RNAse A, and 10 μg/mL PI. The cells were analyzed using a BD FACS Canto II flow cytometer in a PE channel. Ten thousand ‘events’ were acquired per each sample. The data were analyzed using the FACSDiva program (BD Biosci., Franklin Lakes, NJ, USA).

### 4.5. Intracellular Reactive Oxygen Species (ROS) Assay

The intracellular ROS level was measured to detect the anti- or pro-oxidant effect of QU pre-treatment on the SKOV-3/CDDP subline cells. ROS Deep Red dye (RDR; Cellular Reactive Oxygen Species Detection Assay Kit; Abcam, Cambridge, UK) was used to assess the production of ROS. CellROX Deep Red dye freely penetrates into cells, which in turn are oxidized by ROS into a highly fluorescent color. Briefly, the SKOV-3 and SKOV-3/CDDP cells were seeded at a density of 1.5 × 10^5^ cells/well in a 6-well plate, treated with CDDP alone for 4 h or 24 h, QU alone for 24 h or 48 h or followed by CDDP for 4 h or 24 h, and 5 mM H_2_O_2_, as a positive control, (Ecotex, Moscow, Russia; a reference ROS inducer) for 30 min separately. After incubation, the cells were finally stained with 5 µg/mL CellROX Deep Red dye for 40 min at 37 °C in the dark and analyzed on a BD FACSCanto II (BD Biosciences, Franklin Lakes, NJ, USA) flow cytometer in the APC channel (filter 660/20). The data were analyzed using the FACSDiva program (BD Biosciences). The ROS production for each compound was represented as the percentage based on the shift of the RDR fluorescence in the drug-treated vs. untreated control cells. Ten thousand fluorescent ‘events’ were acquired per each sample.

### 4.6. Western Blotting Analysis

Proteins from the treated cells were extracted using RIPA lysis buffer (50 mM Tris-HCl pH 7.4, 1% NP-40, 0.1% sodium dodecylsulfate, 150 mM NaCl, 1 mM EDTA) containing a protein inhibitor cocktail and 2 mM phenylmethylsulfonyl fluoride, then kept on ice for 30 min and centrifuged at 10,000× *g* for 15 min. The total protein in the lysates was quantitated by the Bradford method [41]. The proteins were resolved by 10–15% SDS-PAGE (40 mg total protein/lane) and transferred onto a 0.2 mm nitrocellulose membrane (GE Healthcare Bio-Sci., Chicago, IL, USA). The membranes were blocked with 5% skimmed milk in 1x TBST (Tris-buffered saline with 0.1% Tween-20), incubated for 2 h, and then probed overnight at 4 °C with the indicated primary rabbit antibodies against catalase; mTOR; phospho-mTOR (S^2448^); Trx1; TrxR-1(Abcam (Cambridge, MA, USA)); Trx-2; STAT3; phospho-STAT3 (Tyr^705^ and ser^727^) (Sigma-Aldrich (St. Louis, MO, USA)); caspase 3; caspase 7; caspase 9, PARP; the cleaved caspase 3 (Asp^175^); cleaved caspase 7; cleaved caspase 9 (Asp^330^); cleaved PARP (Asp^214^); and Histon-3/GAPDH (as reference proteins) (Cell Signaling Technology (Danvers, MA, USA)). Next, the membranes were washed twice with 1x TBST and incubated with secondary antirabbit antibodies conjugated with horseradish peroxidase (HRP) (Cell Signaling Tech., Danvers, MA, USA) for 1 h at room temperature. The immunoreactive bands were visualized by enhanced chemiluminescence using the Image Quant LAS 4000 system (GE Healthcare, Chicago, IL, USA).

### 4.7. Apoptotic Programmed Cell Death Analysis

Apoptosis was measured using the Annexin V-FITC apoptosis detection kits (Thermo Fisher Scientific Inc., Waltham, MA, USA). The SKOV-3 and SKOV-3/CDDP cells in the logarithmic growth phase were seeded at a density of 1.5 × 10^5^ cells/well in a 6-well plate, incubated overnight at 37 °C, 5% CO_2_, and treated with CDDP alone or QU alone and followed by cisplatin. The cells were harvested and washed once with PBS and once with 1× binding buffer. The pellets were then resuspended in 100 μL 1× binding buffer, stained with 5 μL of fluorochrome-conjugated Annexin V-APC or FITC, and incubated for 10–15 min at room temperature in the dark. After adding 2 mL of 1× binding buffer, the samples were centrifuged at 400–600 g for 5 min at room temperature. The supernatants were removed; the cells were re-suspended in 200 μL of 1× binding buffer; 5 μL of propidium iodide (PI) staining solution was added and incubated for 5–15 min on ice and analyzed by flow cytometry on a BD FACSCanto II (BD Biosciences, Franklin Lakes, NJ, USA) in the APC channel (filter 660/20 for detection of Annexin V-positive cells) and in the PerCP-Cy™5.5 channel (filter 695/40 for PI-positive cells). The data were analyzed using the FACSDiva program (BD Biosci.).

### 4.8. Statistical Analysis

All the in vitro data presented herein are presented as the mean ± standard deviation (SD). Each of the experiments was repeated at least three times (*n* = 3). Statistical analysis and graphs were generated using GraphPad Prism 8.0 software (San Diego, CA, USA).

## 5. Conclusions

The in vitro data show that polyphenol QU increases the sensitivity of SKOV-3/CDDP cisplatin-resistant human ovarian adenocarcinoma subline cells to cisplatin due to their biological effects, including alteration of the cell cycle, classical pro-oxidation, upregulation of ROS and downregulation of the thioredoxin antioxidant system (Trx/TrxR), and cell signaling (mTOR/STAT3) suppression, which collectively lead to the enhancement of the mitochondrial apoptotic pathway (Cas9, 7, and 3 and PARP). To confirm these results, further studies should be conducted using other cell lines that are resistant not only to cisplatin but also to other chemical drugs; future experiments are needed to validate the effect of QU pre-treatment in vivo to assess their potential to reverse cisplatin resistance in cell lines.

## Figures and Tables

**Figure 1 ijms-24-10960-f001:**
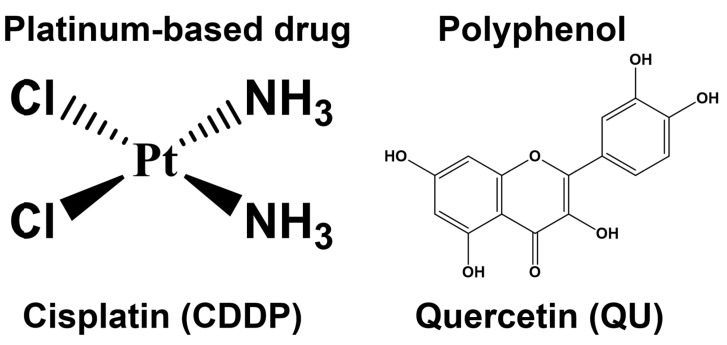
The chemical structures of the platinum-based drug (cisplatin; CDDP) and polyphenol (quercetin; QU).

**Figure 2 ijms-24-10960-f002:**
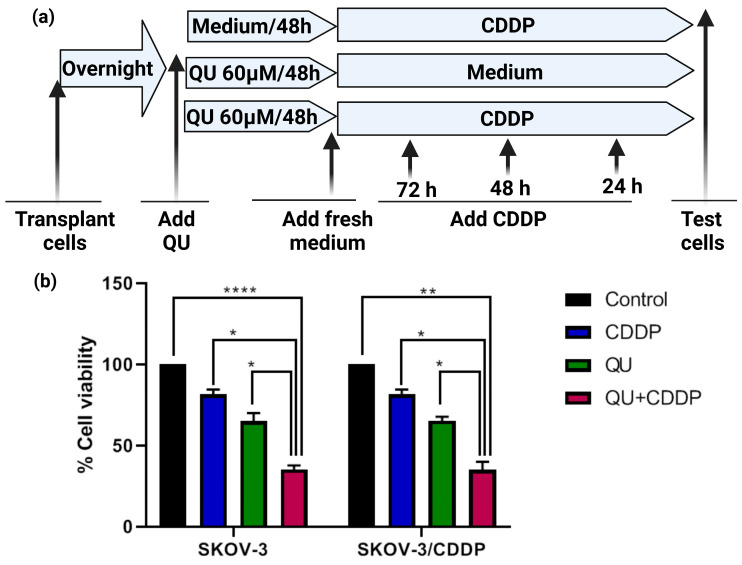
The QU effect on the viability of SKOV3 and SKOV3/CDDP cells. (**a**) Design of QU pre-treatment strategy. Both ovarian cancer cells were either treated individually with CDDP for 24, 48, and 72 h, or with QU for 48 h followed by replacement with fresh medium for an additional 24, 48, and 72 h, or pre-treated with QU for 48 h followed by CDDP for an additional 24, 48, and 72 h after changing medium. (**b**) QU pre-treatment effect on the CDDP efficacy. Both cells were treated with QU (60 µM) for 48 h followed by fresh medium for 72 h and/or CDDP (5 µM for SKOV-3 and 17 µM for SKOV-3/CDDP) for 72 h. Cell viability was assessed using an MTT assay. All data are presented as mean ± SEM and were evaluated using one-way ANOVA followed by Tukey’s corrections for multiple comparisons between different treatments; * *p* = 0.05/0.01, ** *p* = 0.005, **** *p* = 0.0001.

**Figure 3 ijms-24-10960-f003:**
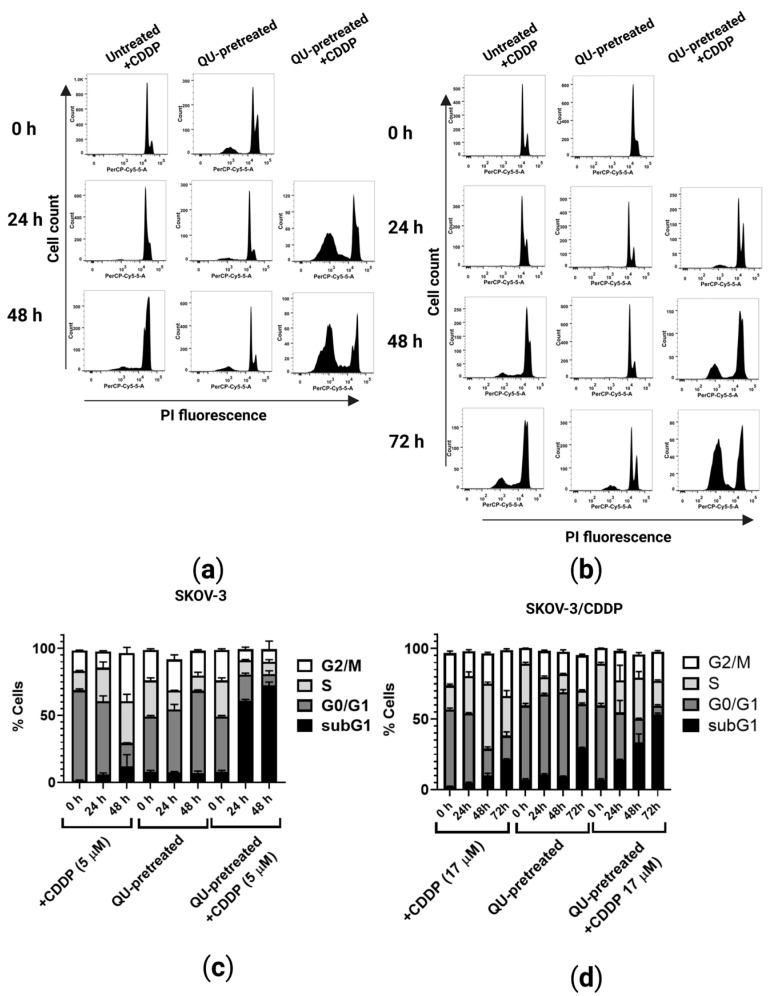
Cell cycle progression of ovarian cancer cells treated with QU and/or CDDP. Cells were incubated with QU (60 µM) for 48 h and then fresh medium and/or CDDP (5 µM for SKOV-3 and 17 µM for SKOV-3/CDDP) were added for further 24, 48, and 72 h incubation. After this exposure period, cell DNA content analysis with PI staining was performed by flow cytometry. (**a**) Representative flow cytometry dot plots; SKOV-3 and (**b**) SKOV-3/CDDP. (**c**) Representative flow cytometry histograms; SKOV-3 and (**d**) SKOV-3/CDDP.

**Figure 4 ijms-24-10960-f004:**
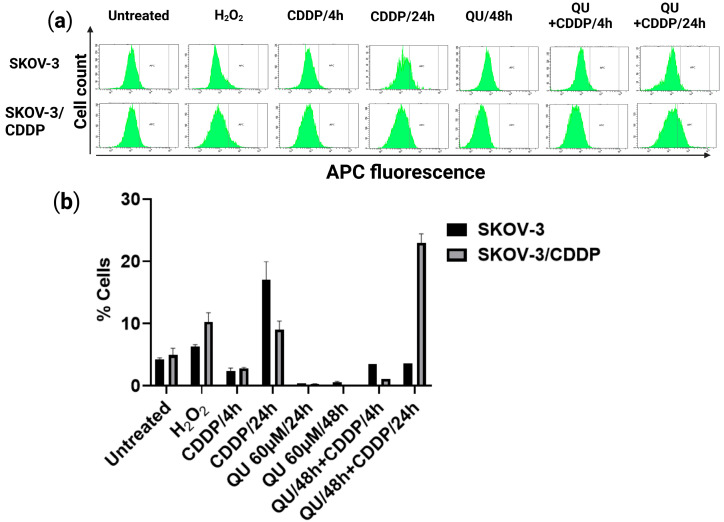
ROS flow cytometry detection in ovarian SKOV-3 and SKOV-3/CDDP cells under QU and CDDP treatment. (**a**) Intracellular ROS level upon either QU or CDDP treatment alone and QU pre-treatment. Both cultured cells were treated with either 60 μM QU (48 h) or 5 μM and 17 μM CDDP (24 h) and QU pre-treatment (48 h), followed by 5μM or 17 μM CDDP (24 h) for SKOV-3 and SKOV-3/CDDP, respectively, and subjected to ROS detection with CellROX Deep Red staining using flow cytometry. (**b**) Quantitative analysis of CellROX Deep Red probe-positive cells.

**Figure 5 ijms-24-10960-f005:**
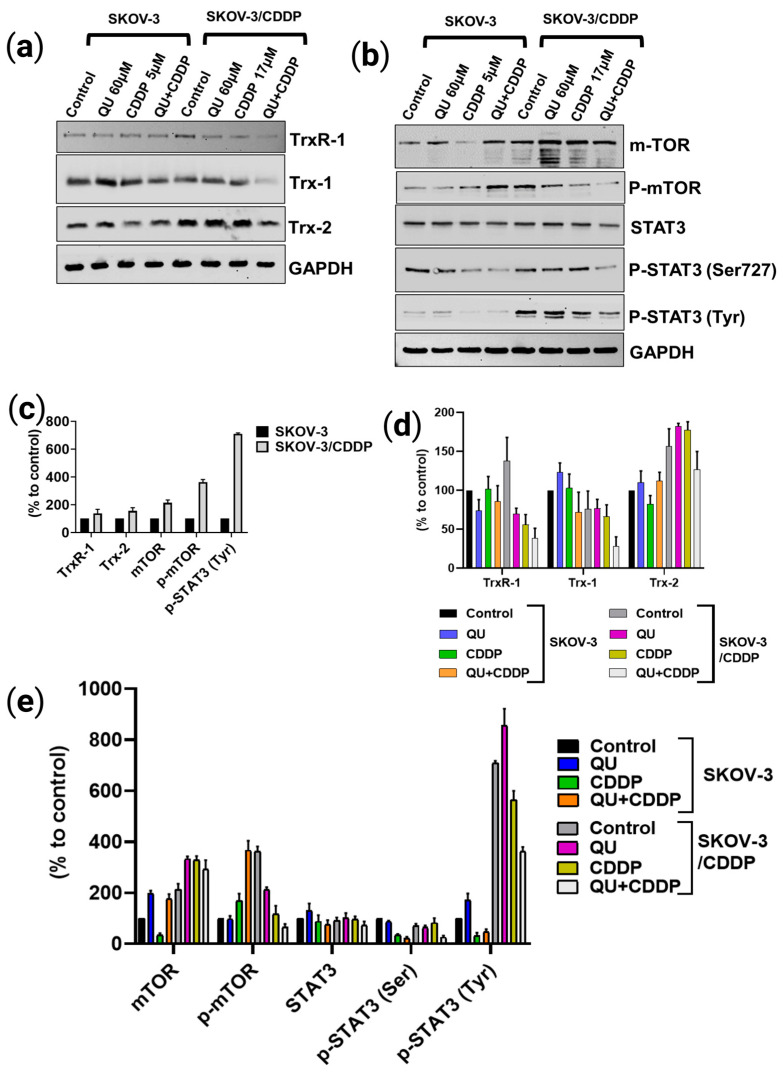
The effect of QU pre-treatment on Trx/TrxR system and mTOR/STAT3 signaling pathway protein expression in ovarian cancer cells using Western blot analysis. Expression of (**a**) Trx/TrxR and (**b**) mTOR/STAT3 protein level in ovarian SKOV-3/CDDP cancer cells compared to SKOV-3 cells. (**c**–**e**) Densitometry results. (**c**) SKOV-3 and SKOV-3/CDDP controls. (**d**) Trx/TrxR system. (**e**) mTOR/STAT3 signaling pathway. The ovarian cancer cells were treated with 60 μM QU (48 h), 5 μM (24 h), or 17μM (48 h) CDDP, or pre-treated with 60 μM QU followed by 5 μM or 17 μM CDDP (24 h and 48 h) for SKOV-3 and (**d**) SKOV-3/CDDP cells, respectively. GAPDH was used as loading control. Values represent mean ± SD (*n* = 3).

**Figure 6 ijms-24-10960-f006:**
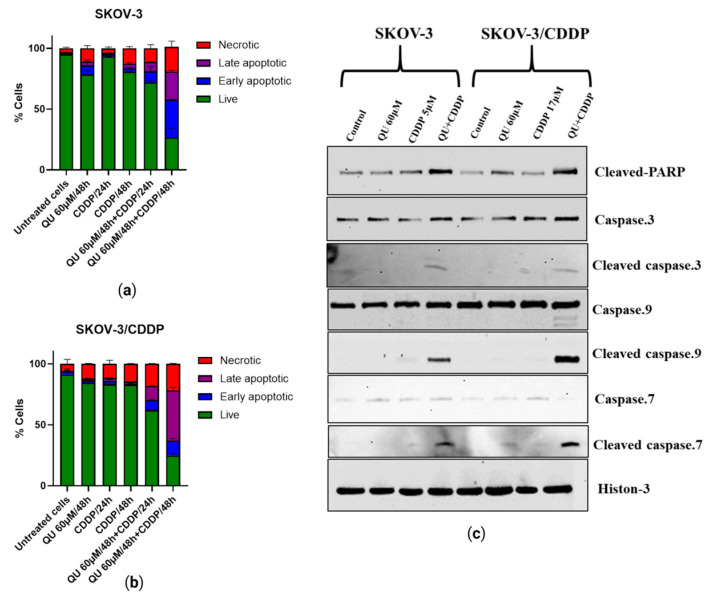
QU-pre-treatment-induced apoptosis in ovarian cancer cells. (**a**) Percent of apoptotic SKOV-3 and (**b**) SKOV-3/CDDP cell lines after treatment with either QU or CDDP and QU pre-treatment for different time periods, 24 and 48 h. (**c**) Representative Western blotting analysis of cleaved caspase-9,7,3 and C-PARP proteins in ovarian cancer cells treated with QU and/or CDDP. Histone was used as loading control. Values represent mean ± SD (*n* = 3).

**Figure 7 ijms-24-10960-f007:**
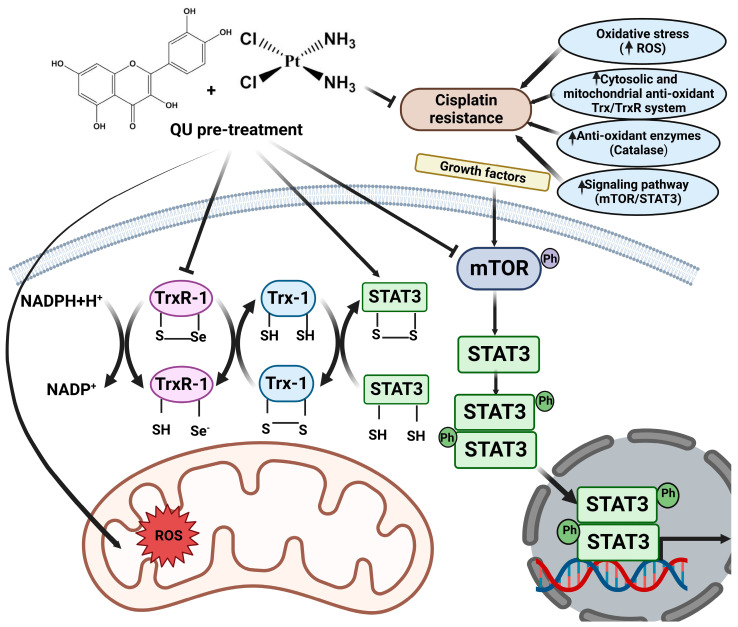
Ovarian SKOV-3/CDDP cells develop resistance to cisplatin through various mechanisms, including oxidative stress, enhanced antioxidant thioredoxin (Trx/TrxR) system, and abnormal signaling pathways. QU pre-treatment has been shown to influence various targets, including Trx/TrxR anti-oxidant system and signaling pathways involved in tumor development and progression. This antioxidant system is composed of thioredoxin reductases (TrxR), thioredoxins (Trx-1), and nicotinamide adenine dinucleotide phosphate (NADPH). Reduced Trx is maintained by TrxR, which accept reducing equivalents from NADPH to the oxidized forms of Trx, that in turn catalyze the reduction of disulfides (s-s) within oxidized cellular proteins, such as (STAT3). Inhibition of TrxR by QU pre-treatment results in the accumulation of oxidized STAT3, which prevents STAT3-dependent transcription. QU pre-treatment also targets mTOR. Inhibition of mTOR is associated with inhibition of phosphorylation of STAT-3 at both Ser^727^ and Tyr^705^ that collectively inhibit SKOV-3/CDDP cell survival and proliferation.

## Data Availability

Not applicable.

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
