# Peer review of "Potentiation of Cisplatin Cytotoxicity in Resistant Ovarian Cancer SKOV3/Cisplatin Cells by Quercetin Pre-Treatment"

_ijms, 2023, doi:10.3390/ijms241310960_

Round 1

Reviewer 1 Report

This is a comprehensive effort. Technically, the manuscript is pretty good, and I have no major concerns. However, I get the feeling they try to downplay the research on quercetin, which is almost exhaustive. It is now clear that quercetin is a potent chemo-preventive molecule with multifaceted effects and intracellular targets. There are a couple of comprehensive reviews that can serve as templates in the introduction section [(Pharmacological basis and new insights of quercetin action in respect to its anti-cancer effects (Si-Min Tang et al. 2020); The Anti-Cancer Effect of Quercetin: Molecular Implications in Cancer Metabolism. Marjorie Reyes-Farias et al. 2019)]. There is a massive number of studies including in-vitro and in-vivo experiments showing that the natural flavonoid has anti-inflammatory, anti-oxidation, and anti-cancer effects. Actions include altering cell cycle progression and promoting apoptosis (the focus of this study) but also inhibiting cell proliferation and angiogenesis, and metastasis progression. Studies do not fall short in relation to ovarian cancer! On this note, there are no studies on ovarian cancer referenced, which is important as the authors claim novelty based on the lack of data on the topic. In fact, a few to mention (D. Catanzaro al. Effect of quercetin on cell cycle and cyclin expression in ovarian carcinoma and osteosarcoma cell lines, 2015; the study suggests that quercetin, exceeding the resistance to CDDP, might become an interesting tool to evaluate cytotoxic activity in combination with chemotherapy drugs. in ovarian carcinoma (SKOV3) human cell lines and in their cisplatin (CDDP)-resistant counterparts (SKOV3/CDDP)- to me that reads as a study akin to the manuscript reviewed here). Other studies have focused on the apoptosis effect (S.H. Kim, et al. Antitumor and apoptotic effects of quercetin on human melanoma cells involving JNK/P38 MAPK signaling activation, 2019; W.J. Lee et al. Quercetin induces mitochondrial-derived apoptosis via reactive oxygen species-mediated ERK activation in HL-60 leukemia cells and xenografts, 2015; X. Gao et al.  Anticancer effect and mechanism of polymer micelle-encapsulated quercetin on ovarian cancer, 2012). In the latter studies, they use A2780S ovarian cancer cell lines to explore the targeting p53 activity in favor of apoptosis, which makes more sense for ovarian cancer as at least 70% of these cancers express p53. The authors need to explain why their study is novel.

Equally, they need to mention potential caveats in the use of this widely available in grapes and vegetables natural molecule; for instance, quercetin has relatively low bioavailability and poor aqueous solubility that potentially hamper its use as a therapeutic agent. The bioavailability of quercetin can be significantly enhanced if it is consumed as a component of whole food.

Minor comments

The previous work needs to be quoted in the Introduction to justify the design of this follow-on study.

A visual abstract of the possible implication of quercetin in the treatment of ovarian cancer is required. 

Minor editing is required. For example, instead of the both, write both

Reviewer 2 Report

1) Figure 2a, why QU 60 uM/48h is duplicated in the figure?

2) Figure 2b, did you test only the IC50 concentration with SKOV3 CDDP? I would have expected to study lower CDDP concentrations tested in the SKOV3 CDDP and a toxic effect after QU treatment to demonstrate the efficacy of the combination treatment. You should include more concentrations.

3) Figure 5. Add error bars

4) This paper only evaluated cell viability, cell cycle and apoptosis. I will suggest that you can present a stronger publication to study the Combination index with the model of Chou–Talay using the CompuSyn software. This can help you to determine synergies between drugs. 

Good English

Round 2

Reviewer 1 Report

You efficiently addressed all comments/suggestions previously made. My only comment remains that the last sentence describing the proposed mechanism of action of QU should come prior to the sentence starting as "taken together...." which should actually be the last sentence concluding the entire concept of the mauscript.

Minor editing required

Author Response

Dear reviewer. We are highly grateful for your consideration of this manuscript and its appreciation. Thank you very much for the valuable comments and suggestions that we used for the correction of the manuscript.

Reviewer 2 Report

All comments were addressed 

Author Response

(The authors gave the same response as above.)
